# The E3 ligase Riplet promotes RIG-I signaling independent of RIG-I oligomerization

Wenshuai Wang [1,2,4], Benjamin Götte [1,4], Rong Guo[3] & Anna Marie Pyle [1,2] ✉

RIG-I is an essential innate immune receptor that responds to infection by RNA viruses. The RIG-I signaling cascade is mediated by a series of post-translational modifications, the most important of which is ubiquitination of the RIG-I Caspase Recruitment Domains (CARDs) by E3 ligase Riplet. This is required for interaction between RIG-I and its downstream adapter protein MAVS, but the mechanism of action remains unclear. Here we show that Riplet is required for RIG-I signaling in the presence of both short and long dsRNAs, establishing that Riplet activation does not depend upon RIG-I filament formation on long dsRNAs. Likewise, quantitative Riplet-RIG-I affinity measurements establish that Riplet interacts with RIG-I regardless of whether the receptor is bound to RNA. To understand this, we solved high-resolution cryo-EM structures of RIG-I/RNA/Riplet complexes, revealing molecular interfaces that control Riplet-mediated activation and enabling the formulation of a unified model for the role of Riplet in signaling.

The innate immune receptor RIG-I (Retinoic acid inducible gene-I) is an indispensable first-line defender against infection by RNA viruses, including coronavirus and influenza viruses that have caused severe pandemics[1–7]. It is critical that we understand molecular mechanisms of RIG-I activation in order to develop new therapeutic strategies against viral infection, cancer, and other disorders of immune function.

Post-Translational Modifications (PTMs) are essential during the process of RIG-I signaling, where they act by repressing or activating the RIG-I sensor[1]. The most well-characterized example is the K63-linked ubiquitination of RIG-I signaling domains (CARDs, caspase recruitment domains), which enhances RIG-I signaling by stabilizing RIG-I in an active conformation[1,8,9]. When RIG-I forms a complex with viral RNAs in the cytosol, it undergoes a structural change that exposes its signaling domains (CARDs)[1,10], resulting in an activated conformation that is modified by covalent conjugation with K63-Ubiquitin chains. The K63-Ub chains are thought to help bring together CARDs from different activated RIG-I molecules, thereby forming a stable CARDs tetramer[11] that binds to the MAVS adapter protein (Mitochondrial antiviral-signaling protein), thereby inducing MAVS aggregation on mitochondria via RIG-I-CARDs:MAVS-CARD interactions[9], thereby

leading to IFN (interferon) activation and a rapid antiviral signaling cascade.

Years of work in many labs have established that the E3 ligase Riplet (RING finger protein leading to RIG-I activation) activates RIG-I signaling by conjugating K63-Ub chains to RIG-I CARDs[12–15]. These studies have shown that Riplet is not only a ligase, but that it also forms a specific complex with RIG-I. These previous studies utilized long RNA molecules to stimulate the RIG-I receptor, which has resulted in a model for Riplet-mediated RIG-I activation that rests on two concepts: 1. A requirement for Riplet to interact with RIG-I that has oligomerized as a filament on long dsRNA molecules. 2. Given that Riplet forms a homodimer, it was proposed to act as a physical bridge between RIG-I residues on different RIG-I/RNA filaments, thereby forming giant plaques that stimulate signaling. While these aspects of the Riplet model were reasonable given the information available at the time, new information on the RIG-I signaling mechanism has made it important to re-evaluate the physical basis for Riplet function. For example, it has been unambiguously shown that RIG-I forms functional signaling complexes on short RNA duplexes that are unable to form filaments[16–18]. In addition, cell biology and imaging have now established that, during the timeframe of signaling, RIG-I does not form

[1]Department of Molecular, Cellular and Developmental Biology, Yale University, New Haven, CT 06511, USA. [2]Howard Hughes Medical Institute, Yale University, New Haven, CT 06520, USA. [3]Department of Chemistry, Yale University, New Haven, CT 06511, USA. [4]These authors contributed equally: Wenshuai Wang, Benjamin Götte. ✉e-mail: anna.pyle@yale.edu

massive aggregated complexes or plaques[19,20]. There is no experimental evidence for the formation of tangled filaments in the mechanism of RIG-I signaling, and it has been established that RIG-I readily signals without them[18,19,21]. Given these issues, we wanted to take a fresh look at the physical structure of functional Riplet/RIG-I complexes in order to better understand the molecular basis for signaling.

In this work, to better understand the molecular basis for Riplet recognition of RIG-I/RNA complexes, we present a series of Riplet-RIG-I complexes and examine Riplet-mediated RIG-I activation by different types of RNA ligands. The resulting data establish that Riplet activates RIG-I in complex with very short dsRNAs, thereby negating the requirement for RNA-based filaments in the mechanism of RIG-I activation. Intriguingly, Riplet forms RIG-I complexes with moderate affinity, with and without RNA, suggesting that Riplet is constitutively associated with RIG-I in the cytoplasm. We developed a pipeline for determining single particle cryo-EM structures of RIG-I:RNA:Riplet ternary complexes using full-length proteins. This work reveals that Riplet recognizes a relatively small, rigid region on the surface of RIG-I, thereby providing a structural explanation for Riplet affinity for both apo and RNA-bound RIG-I. By combining this data with mutational functional analysis and AlphaFold predictions of unresolved Riplet domains, we develop a unified model for RIG-I activation by E3 ligase Riplet.

## Results
### Riplet regulates RIG-I activation without forming RIG-I filaments
To explore whether Riplet facilitates RIG-I activation in the absence of filament formation, we stimulated RIG-I with a short 5′-triphosphorylated 14-basepaired stem-loop RNA (p3SLR14, Supplementary Fig. 1) which can bind only one RIG-I molecule, and which is well established as a potent RIG-I agonist (Fig. 1a)[18,19]. Indeed, p3SLR14 induces IFN responses comparable to long dsRNAs from polyinosinic:polycytidylic acid [poly(I:C)] and Sendai virus (SeV)[18,19]. Consistent with this, p3SLR14 stimulated IFN activation at similar level to long RNA molecules (p3SLR50, 5′-triphosphorylated 50-basepaired stem-loop RNA; Fig. 1a), confirming that the short RNA is sufficient to activate IFN response. We tested the resulting impact of Riplet on IFN activation using p3SLR14 in order to determine if it induced a response comparable to that of Riplet on long dsRNA-stimulated IFN activation[12–15]. To this end, we generated a Riplet knock-out (KO) HEK293T cell line (Supplementary Fig. 2). As observed previously in the presence of long dsRNAs, overexpressed Riplet in parental HEK293T cells significantly boosted the IFN response (Fig. 1b), and it consistently rescued the abolished IFN response in Riplet KO cells (Fig. 1c). These findings establish that Riplet plays a role in RIG-I activation even when RIG-I is bound to short dsRNA ligands and that its activity is independent of RNA filament formation.

### Riplet activates RIG-I mediated by RING and PrySpry domains
We next explored the role of individual Riplet domains on RIG-I activation by short RNA ligands. Riplet is a homodimer composed of an N-terminal RING domain (catalytic domain), CC domain (coiled-coil domain, homodimerization), and C-terminal PrySpry domain (RIG-I recognition domain, Fig. 1d). To examine the role of each domain, we created a series of Riplet truncations and tested their impact on IFN activation. The deletion of either RING or PrySpry domains abolished the IFN response, and neither the RING nor PrySpry domains alone can activate IFN response (Fig. 1e). Thus, both the RING and PrySpry domains are critical for RIG-I activation and they must function in cis, where PrySpry functions to recognize RIG-I and RING ubiquitinates RIG-I.

Although the CC domain appears to stabilize the Riplet homodimer and was proposed to activate the IFN response by bridging RIG-I molecules located on different strands of long dsRNA[14], it had not been established that this domain is strictly required for RIG-I signaling, and its role had not been tested when RIG-I is bound to short dsRNA agonists. To address these issues, we removed the CC domain (120–183) from Riplet by replacing it with a set of flexible GS (Gly-Ser) linkers (GS repeats with 52 and 14 amino acids). Surprisingly, in the cell-based IFN reporter assay, we found that replacement of the CC domain still induced comparable IFN response to that of wildtype Riplet (Fig. 1e). Consistent with this behavior, a titration of Riplet expression and p3SLR14 revealed similar patterns of IFN response for both WT Riplet and CC-removed mutants (Fig. 1f, g), despite the fact that only RING and PrySpry domains are retained in the mutant construct. This means that Riplet mutants that maintain the covalent continuity between RING and PrySpry domains, even without the features provided by the CC domain, are sufficient to activate RIG-I and that they can achieve this even in the presence of a minimal RIG-I agonist that is incapable of forming filaments. That said, it remains possible, and even likely, that CC domain-mediated dimerization plays an as-yet uncharacterized role under other types of RIG-I activation conditions or viral infections.

### Riplet activates RIG-I, despite recognizing RIG-I at low affinity
Having confirmed that Riplet can activate p3SLR14-mediated RIG-I signaling in cellulo, we next evaluated whether these components can form a stable complex in vitro, as direct complex formation has never been investigated previously using full-length, unconjugated proteins. We employed Surface Plasmon Resonance (SPR) to measure the binding affinity ($K_D$) of different Riplet constructs to RIG-I with and without RNA, revealing the kinetic and thermodynamic parameters governing the association between Riplet and RIG-I. Surprisingly, full-length Riplet recognizes both apo-RIG-I and RIG-I:p3SLR14 complex with similar, moderate affinity (0.7 μM, Fig. 2a and Supplementary Fig. 3a), suggesting that Riplet does not distinguish p3SLR14-bound from unbound RIG-I. Although Riplet mainly utilizes the PrySpry domain to recognize RIG-I, the Riplet RING domain is expected to bind directly to RIG-I as well because it is the enzyme that covalently attaches ubiquitin to RIG-I CARDs. Consistent with this, deletion of the RING domain slightly but reproducibly reduces the affinity of Riplet to RIG-I:p3SLR14 (Fig. 2a and Supplementary Fig. 3a, 0.7– 1.2 μM). Furthermore, the PrySpry domain alone recognizes RIG-I with slightly lower affinity than full-length Riplet (Fig. 2a and Supplementary Fig. 3a).

In the context of p3dsRNA24-bound RIG-I, a complex that contains two RIG-I molecules and hence, a 2:2 stoichiometry for the Riplet homodimer binding sites, the binding affinity with Riplet was about 100-fold higher (6.6 nM) compared to p3SLR14-bound RIG-I (0.7 μM). Whereas deletion of the RING only moderately affected the binding affinity of Riplet to p3dsRNA24:RIG-I (10 nM), deletion of both, RING and CC domains, severely reduced the binding affinity to 1.4 μM.

These data indicate that RIG-I recognition is primarily led by the PrySpry domain, supported by the CC domain, and that the RING domain has a small influence on RIG-I recognition.

### Overall structures of RIG-I:RNA:Riplet ternary complex
Having established that Riplet forms a direct complex with full-length RIG-I, establishing a pipeline for cryo-EM structure determination became feasible. To that end, we attempted to isolate RIG-I:SLR14:Riplet complexes in vitro. However, electrophoretic mobility shift assays (EMSA) indicated that this complex was just below the level of stability needed to obtain a complex under cryo-EM conditions (Fig. 2b), which is consistent with the moderate binding affinity of Riplet under these conditions (0.7 μM). Given that Riplet is a stable homodimer, we reasoned that it might be possible to anchor it more firmly onto the RNA ligand through bidentate coordination with two RIG-I molecules that are bound at a fixed distance in space through high-affinity interactions with two blunt 5′-triphosphorylated dsRNA termini (~0.4 nM affinity for RIG-I to this type of RNA site). Given that the footprint of

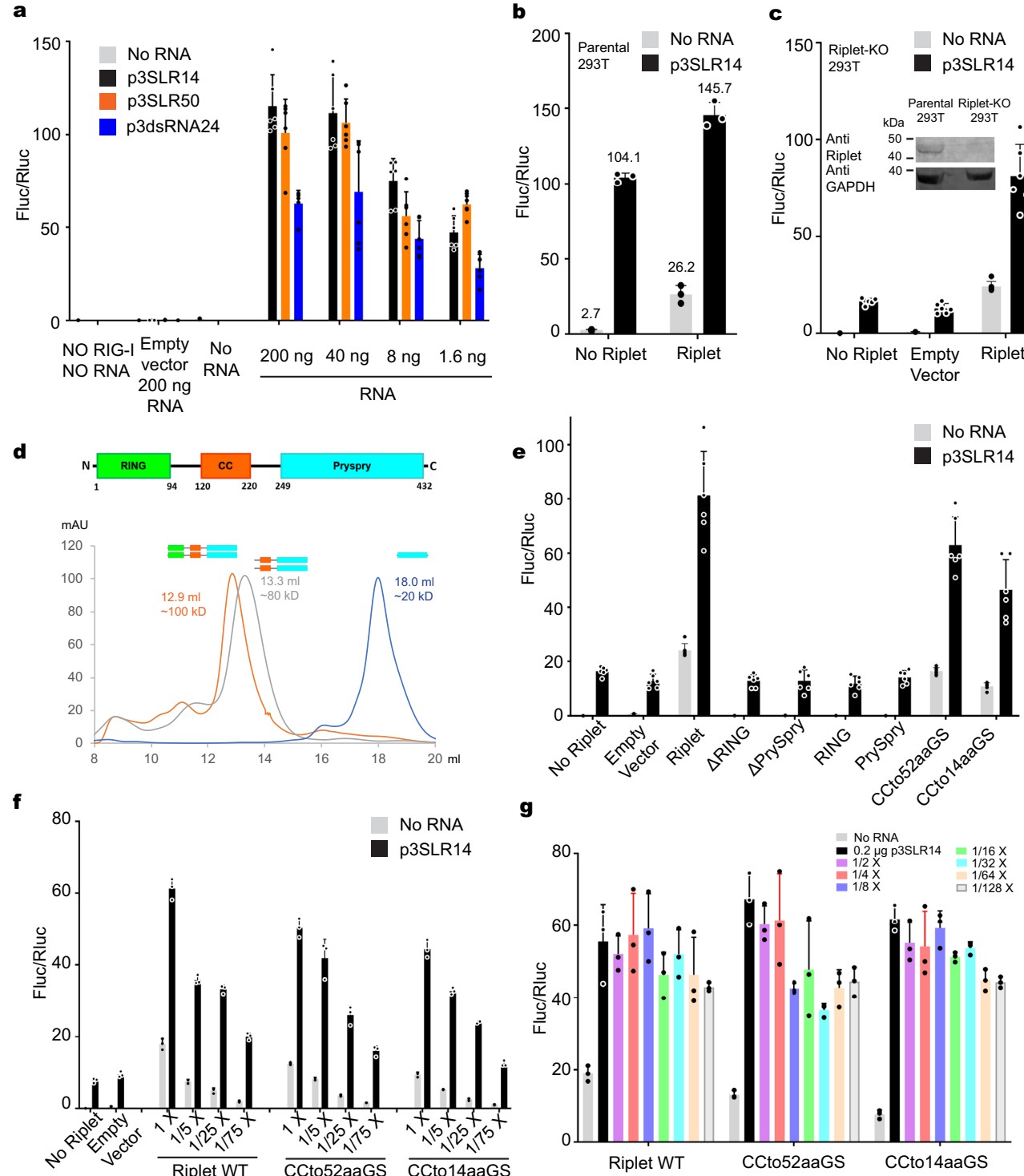

**Fig. 1 | Riplet is required for RIG-I signaling in the presence of short dsRNA p3SLR14.** **a** IFN-β induction by transfection of RIG-I stimulated with p3SLR14, p3SLR50 and p3dsRNA24 in parental HEK293T Cells (*n* = 6 biological replicates). **b**, **c** IFN-β induction by co-transfection of RIG-I and Riplet stimulated with p3SLR14 in parental (**b**; *n* = 3 biological replicates) and Riplet-KO (**c**; *n* = 6 biological replicates) HEK293T Cells. The expression of Riplet is illustrated. **d** Schematic representation of the Riplet protein. Starting from the N-terminus, the RING, CC and PrySpry domains are in green, orange, and cyan, respectively. The elution volumes of Riplet WT and truncation mutants by gel filtration are rendered accordingly. The

mAU (y-axis) means milli absorbance units. **e** IFN-β induction by co-transfection of RIG-I and Riplet truncation mutants stimulated with p3SLR14 in Riplet-KO HEK293T Cells (*n* = 6 biological replicates). **f** IFN-β induction by co-transfection of RIG-I and titrated Riplet CC mutants stimulated with p3SLR14 in Riplet-KO HEK293T Cells (*n* = 3 biological replicates). **g** IFN-β induction by co-transfection of RIG-I and Riplet CC mutants stimulated with titrated p3SLR14 in Riplet-KO HEK293T Cells (*n* = 3 biological replicates). Data are represented as mean ± SD. Source data are provided as a Source Data file.

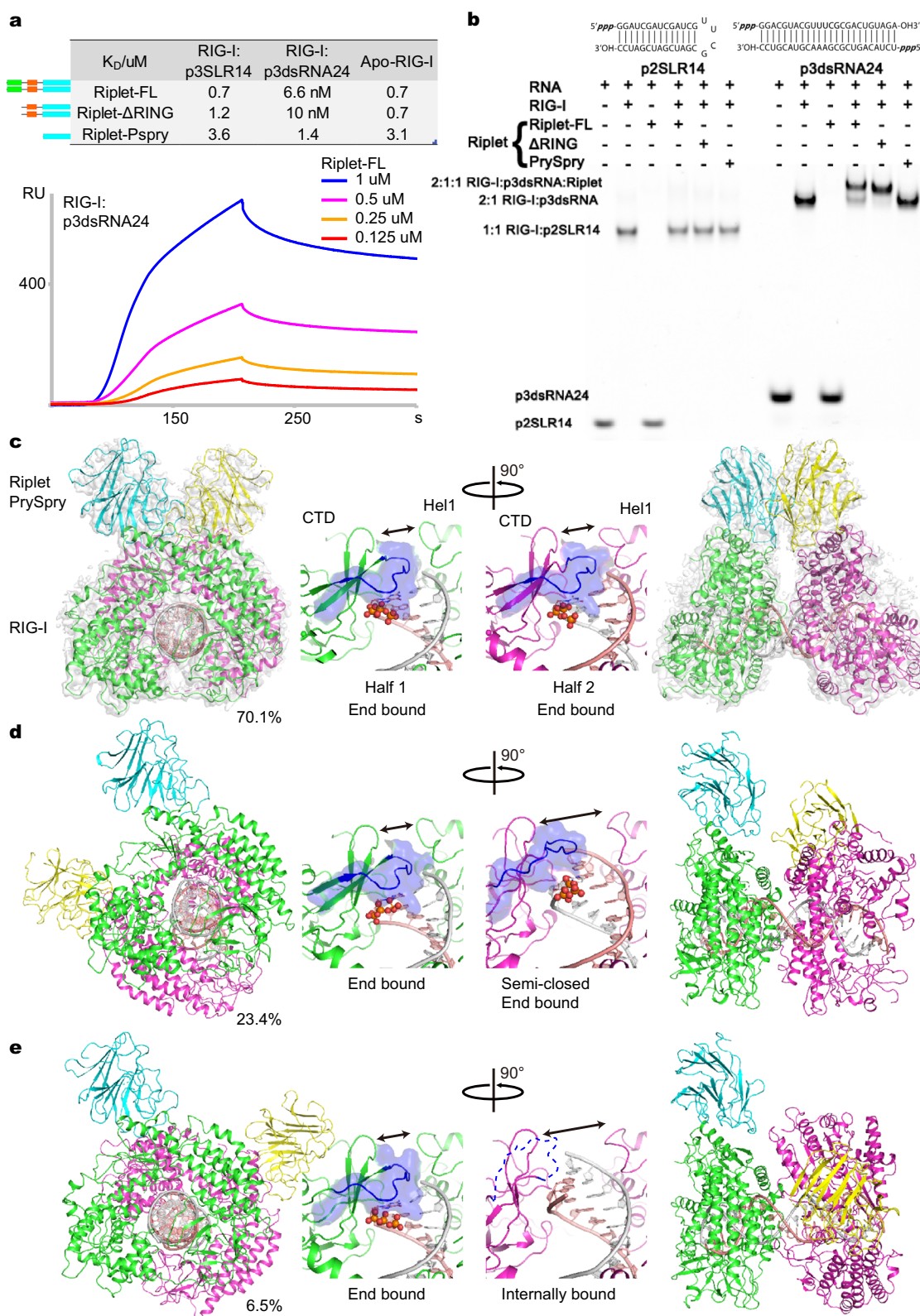

**Fig. 2 | Overall structure of the RIG-I:p3dsRNA24:Riplet complex. a** Binding affinity characterization of Riplet to RIG-I:p3SLR14, RIG-I:p3dsRNA24 and apo RIG-I by SPR. The binding curves of Riplet-FL to RIG-I:p3dsRNA24 are rendered. **b** Electrophoretic mobility shift assay (EMSA) of Riplet binding to RIG-I in the presence of different RNAs. This EMSA was repeated independently twice with similar results. **c–e** Overall structures and zoom-in views of CTD binding pockets for the end-end complex (**c**), end-semi-closed end complex (**d**), and end-inter complex (**e**). RIG-I from half 1 and half 2 is in green and magenta, respectively. Riplet from half 1 and half 2 are in cyan and yellow, respectively. Residues involved in recognizing the RNA terminus are rendered as cartoons and surface in blue. The phosphate groups are rendered as spheres. Source data are provided as a Source Data file.

RIG-I on dsRNA is ~10 bp, we designed a 24-basepaired dsRNA (p3dsRNA24) bearing two 5′-triphosphorylated blunt dsRNA termini for recruiting two RIG-I molecules (Supplementary Fig. 1). Given the entropic advantage of the chelate effect, Riplet is expected to form a more stable complex with a 2RIG-I-p3dsRNA24 complex. Indeed, p3dsRNA24 successfully forms a stable ternary complex, resulting in a significantly enhanced Riplet $K_D$ (from 700 nM to 6.6 nM, Fig. 2a, b), enabling large-scale purification of the complex for subsequent structural work.

Employing the sample preparation and image processing pipeline established previously[21,22], we obtained three different structures of the bidentate RIG-I:p3dsRNA24:Riplet complex, solved at 3.2, 4.0, and 3.9 Å (Supplementary Fig. 4). While previous structural studies used artificially fused Riplet[PrySpry]-RIG-I[ΔCARDs] [23], here we visualize the non-covalently assembled RIG-I:RNA:Riplet ternary complex, containing full-length proteins. Unfortunately, only the structure of the PrySpry domain of Riplet but not the RING or CC domain could be resolved. Each structure contains two RIG-I molecules, one p3dsRNA24, and one Riplet homodimer, but they differ in the relative positions of two PrySpry domains, which depend on the relative positions of their partner RIG-I molecules (Fig. 2c–e). In the structure reconstructed from the major population of particles (70.1%), each triphosphorylated terminus is bound to an individual RIG-I molecule (end-bound complex), thereby enabling the Riplet proteins to adopt a head-to-head arrangement. In this case, each PrySpry domain recognizes a specific motif on the outer surface of RIG-I Hel2 (Fig. 2c).

In the structure containing 23.4% particles, one RIG-I forms a typical end-bound complex with the terminal triphosphate, but the second RIG-I binds to the opposite dsRNA terminus in a "semi-closed" end-bound state, where the CTD has tilted away from its normal orientation while maintaining engagement with the terminal dsRNA diphosphate (Fig. 2d). This observation is consistent with RIG-I/RNA recognition models in which CTD:RNA interactions are the first point of contact upon RIG-I encounter with RNA, and it provides new insights into the conformational variability of RIG-I CTD complexes.

In the structure composed of the smallest population of particles (6.5%), the second RIG-I is not bound at the opposite terminus at all, but instead binds to a proximal internal duplex site. This results in a head-to-tail arrangement between bound Riplet molecules (Fig. 2e). Like other internally-bound structures of RIG-I, the CTD is tilted and its conserved diphosphate recognition pocket has melted to accommodate the RNA backbone (Fig. 2e). This is an interesting result because it suggests that Riplet forms stabilized complexes in cases where two RIG-I molecules are bound on the same RNA, even if there is only one triphosphate end (such as a blunt dsRNA replication intermediate > 20 bp in size). Importantly, we observe that the relative position between the RIG-I molecules is not very important due to flexibility of the Riplet linkers (Fig. 2e). There may be cases where this bidentate Riplet association is advantageous for effective signaling. That said, the p3dsRNA24 ligand does not yield a higher IFN response than p3SLR14 complexes that are half that size (Fig. 1a), indicating that bridging by Riplet is unlikely to be strictly required for RIG-I-mediated IFN activation.

As the second RIG-I in each structure recognizes different types of RNA sites, the two PrySpry domains bind to RIG-I from different approach angles, revealing variable distances between them, ranging from 58 Å to 101 Å (Supplementary Fig. 5). This finding suggests that the CC domains in these complexes are more dynamic than expected (consistent with our inability to visualize them), enabling the complex to accommodate different spacings between sites of PrySpry domains, and enabling PrySpry domains to approach RIG-I Hel2 from different angles. This means that Riplet can facilitate RIG-I recognition of many different types of RNAs, forming strong bidentate complexes with a diversity of species.

## Characteristics of RIG-I recognition by the Riplet PrySpry domain

Despite the different relative orientations of PrySpry in the three structures, the PrySpry domain uses the same interface with RIG-I Hel2 in each case (Supplementary Fig. 6). Using the major particle class as an example, the interacting interface buries a comparatively small solvent-accessible surface area (889.5 Å$^2$ on average, 1200–1700 Å$^2$ for antigen-antibody complex), consistent with the low affinity between Riplet and apo-RIG-I/RIG-I:p3SLR14 (0.7 μM). At this interface, the Riplet PrySpry domain uses several loops to recognize a rigid motif composed of an α-helix in the outer surface of the RIG-I Hel2 domain (Fig. 3a). The side chains of W415, Y417, and L419 in PrySpry insert into the hydrophobic pockets formed by F616, I617 and L624, while R342 and E350 in PrySpry make polar contacts with RIG-I Hel2 (Fig. 3b), resulting in an interface that is consistent with the first RIG-I:Riplet structures obtained using artificially fused Riplet[PrySpry]-RIG-I[ΔCARDs] [23]. Most of the key residues at the interface of RIG-I and Riplet are highly conserved among mammals (Fig. 3c, e). Single point mutations in Riplet, including F616A, I617A, and L624A in RIG-I and W415A, Y417A, and L419D, which are located at the RIG-I:Riplet interface, drastically diminished the IFN response in the context of RNA ligand p3SLR14 (Fig. 3d, f). This finding confirms that Riplet interacts with RIG-I:p3SLR14 under physiological conditions in the same way as that illustrated by the structural work using p3dsRNA24. These findings underscore the importance of PrySpry:Hel2 interactions for successful Ub conjugation and eventual IFN activation, even in the presence of short dsRNA agonists.

## Implications of the predicted dimeric-Riplet structure on RIG-I activation

While the CC domain was previously thought to play a primary role in Riplet dimerization (Fig. 1d)[14], it is now clear that RING domain dimerization is the prevailing determinant for conferring E3 ligase activity[24]. We therefore sought to investigate the structural basis for RING and CC dimerization-mediated RIG-I ubiquitination using other methods, particularly given the inability to visualize them by cryo-EM in this study (Fig. 2c). To this end, we modeled the tertiary structures of Riplet RING, CC domain, and full-length Riplet dimers using ColabFold, which can predict structures of complexes (Fig. 4a)[25]. Consistently, both RING and CC domains were predicted to form an extended dimer with confidence above 90, with high confidence and low PAE (Predicted Aligned Error) in the dimeric interface of the RING and CC domains (Fig. 4b). Similar to the solved structures of RING domains from Riplet homologs (such as Trim25), Riplet RING dimerization is maintained by a 4-helix bundle that is formed by the N-terminal and C-terminal α-helices of each RING domain (Fig. 4c and Supplementary Fig. 7a, b). A unique difference in the Riplet case is that the N-terminus of two Riplet RINGs fold into β-strands that assemble to an antiparallel β-sheet, which stabilizes the 4-helix bundle through extensive hydrophobic interaction networks and a larger buried surface area than Riplet homologs (Fig. 4c, e and Supplementary Fig. 8; Riplet RING vs Trim25 RING, 1500 Å$^2$ vs 1000 Å$^2$). These findings suggest that Riplet RING might form a dimer to perform Ub conjugation like other TRIM E3 ligases under physiological conditions[26,27] and it may explain why the CC-removal mutant still activates the IFN response (Fig. 1e). These findings suggest that Riplet is uniquely dependent on the RING domain, which may have ramifications for the activation mechanism.

According to the predicted structure of the CC domain, the Riplet CC domain contains a shorter coil-coiled region than Trim25, and folds into parallel coiled coils unlike the antiparallel coiled coils in Trim25 (Fig. 4b, d)[28]. The conserved hydrophobic residues projecting from the α-helices of each CC domain establish an interaction network, thereby maintaining the dimeric conformation (Fig. 4e and Supplementary Fig. 8). In addition, the predicted structure of the PrySpry domain closely resembles the resolved cryo-EM structure, adding confidence

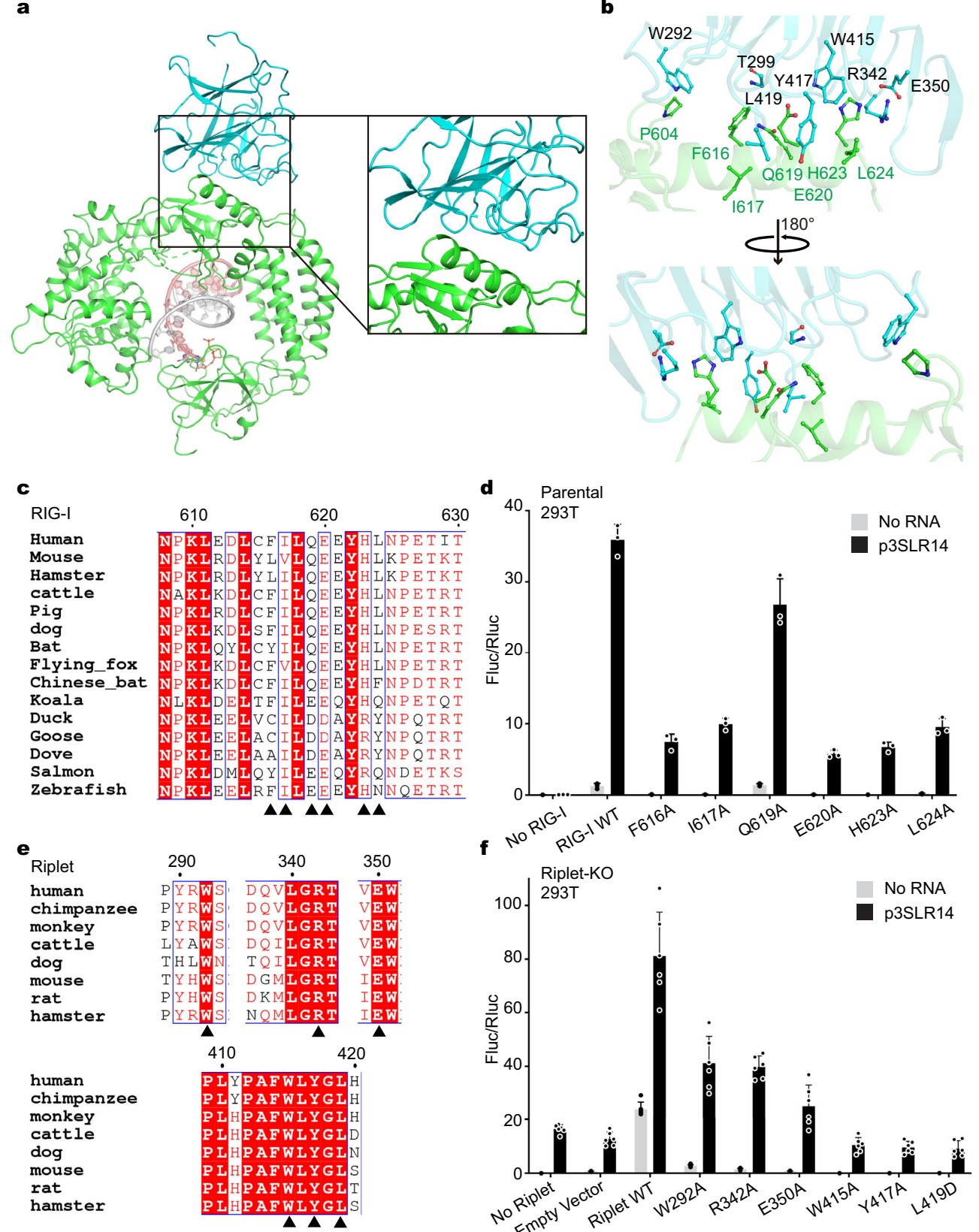

**Fig. 3 | Characteristics of the Riplet-RIG-I recognition interface. a** Overview of the Riplet:RIG-I interface. Riplet PrySpry domain in cyan and RIG-I in green. **b** Interactions between Riplet PrySpry domain (cyan) and RIG-I Hel2 domain (green). The key residues are rendered as sticks. **c** The conservation of residues in RIG-I Hel2 domain participating in Riplet interactions. The key residues are indicated with black triangles. **d** IFN-β induction by RIG-I mutants stimulated with p3SLR14 in parental HEK293T Cells (*n* = 3 biological replicates). **e** The conservation of residues in the Riplet PrySpry domain that participate in RIG-I interactions. The key residues are denoted with black triangles. **f** IFN-β induction by RIG-I and Riplet mutants stimulated with p3SLR14 in Riplet-KO HEK293T Cells (*n* = 6 biological replicates). Data are represented as mean ± SD. Source data are provided as a Source Data file.

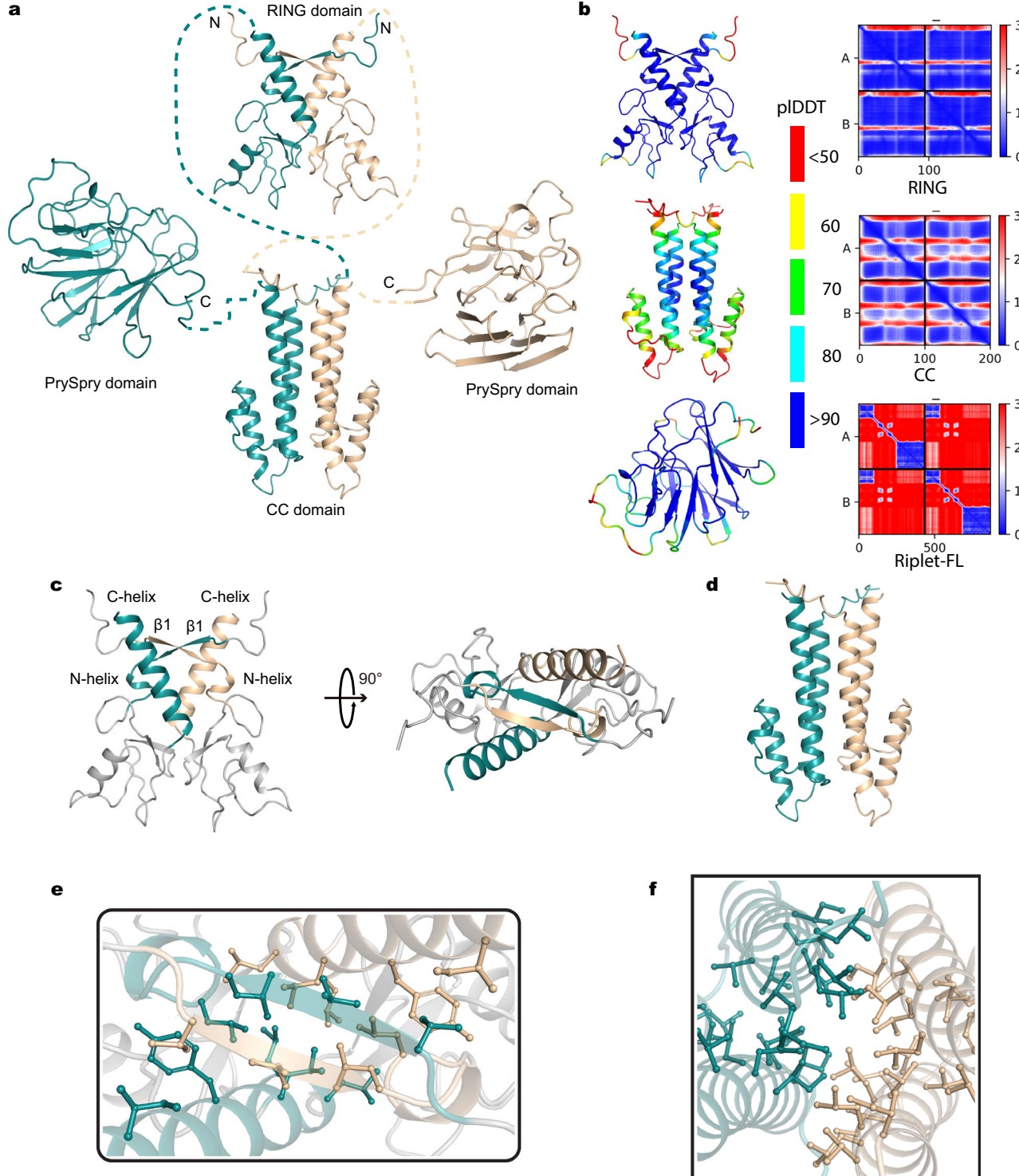

**Fig. 4 | Predicted structure of Riplet. a** Overall structure of Riplet predicted by ColabFold. Two Riplet molecules are colored in deepteal and wheat, respectively. **b** Confidence map and PAE (Predicted Aligned Error) plot of Riplet domains. Residues with pLDDT (predicted local distance difference test) are colored in red (<50), yellow (60), green (70), cyan (80) and blue (>90). **c** Overall structures of the Riplet RING dimer. The N-terminal, C-terminal helices and β-sheet are colored. **d** Overall structure of the CC domain. **e, f** Zoom-in views of the dimeric interface of RING (**e**) and CC (**f**) domains. The hydrophobic residues at the interface are rendered as sticks.

to the computational modeling approach (Supplementary Fig. 7b). By ultimately combining predicted structures of apo Riplet domains and solved structures of the RIG-I:RNA:Riplet complex, we successfully created a model of the entire dimeric protein/RNA complex, where the RING, CC, and PrySpry domains are connected by flexible linkers (Fig. 5a). This model for the intact complex provides insights into the

working mechanisms for RIG-I recognition and ubiquitination by Riplet (vide infra).

## Discussion

In this work, we show that the Riplet E3 ligase establishes a conserved interface with the RIG-I motor domain and we confirm that this

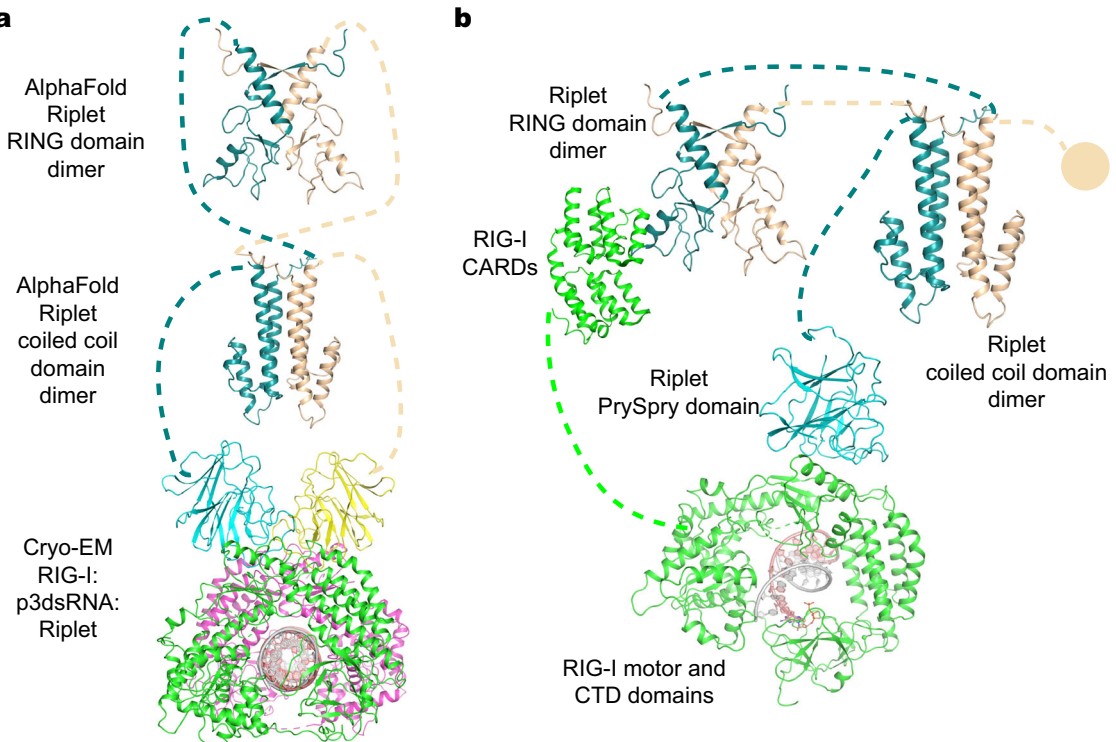

**Fig. 5 | Model of RIG-I activation by Riplet. a** Composite structure of RIG-I recognition by full-length Riplet. The predicted structures of RING and CC are used to represent the domains missing from the cryo-EM structure of RIG-I:p3dsRNA24:Riplet complex. Two RIG-I molecules are colored in green and magenta, while the predicted and resolved domains of two Riplet in deepteal/wheat and cyan/yellow, respectively. **b** Working model of RIG-I activation by Riplet. Riplet is recruited to viral RNA-bound RIG-I by Riplet-PrySpry:RIG-I-Hel2 interactions, followed by Ub conjugation to RIG-I-CARDs mediated by the dimeric Riplet-RING domain. The second PrySpry domain could be in any of the three states: unbound form①, bound to apo RIG-I②, bound to RIG-I:RNA complex③, which is not expected to affect the process of Ub conjugation to the first RIG-I molecule.

interface is required for positioning the ligase ubiquitination of the RIG-I CARDs, thereby transducing the signaling pathway. We show that Riplet establishes interaction with RIG-I in many different structural contexts, including those in which RNA is not bound at all, which has important implications for the mechanism of signaling in healthy and dysregulated cellular states. By solving high-resolution structures of intact RNA/RIG-I/Riplet complexes, we revealed the molecular interfaces among these components and established mechanistic roles for the structural domains created through Riplet dimerization. These findings shed new light on the sequential cascade of events that initiate RIG-I signaling.

One of the most mechanistically significant findings in our study is that Riplet is constitutively bound to RIG-I, forming complexes with the receptor in the presence or absence of RNA. Riplet binds to both apo and RNA-bound RIG-I molecules, showing the same affinity for the apo and p3SLR14-bound RIG-I. This means that Riplet is an integral part of the RIG-I machinery, and that it is not recruited upon RNA binding, in contrast to conventional schemes for RIG-I activation by the ligase. This result should not have been surprising, as the PrySpry interaction domain binds on the surface of RIG-I Hel2, far from the RNA binding interface. It will have equivalent affinity for RIG-I whether RNA is bound or not, which is clear from the direct binding studies reported here. The mechanism by which Riplet selectively ubiquitinates the CARDs of RIG-I molecules during an antiviral response is therefore dictated by sustained presentation of the CARDs that occurs when RIG-I is bound to dsRNAs terminated by 5′-diphosphate groups. It is now established that viral RNA binding traps RIG-I in a highly activated, CARDS-out conformation, which makes the CARDS readily accessible for attack by the flexibly tethered, adjacent RING domains[21]. These findings are consistent with the observation that overexpression of Riplet can trigger RIG-I signaling under conditions where every expressed RIG-I

molecule is bound to Riplet, even a fleeting sampling of the CARDs-out conformation will lead to ubiquitination and signaling, even in the absence of RNA.

The fact that SLR14 stimulates Riplet-mediated signaling, despite the fact that it contains only one binding site for a RIG-I molecule, means that Riplet function does not require RIG-I to be oligomerized or filamentized along a large RNA. Riplet can ubiquitinate a single RIG-I molecule that is bound at a single RNA terminus and still relay a functional IFN activation signal. The fact that SLR14 and dsRNA (p3dsRNA24) induce the same amount of Riplet-mediated RIG-I signaling indicates that there is no apparent mechanistic benefit for Riplet to associate with multiple RIG-I units that are bound on longer RNAs. That said, the results we report here were obtained in a reporter construct that may not reflect the activation constraints that are seen in all cell types or in all types of viral infection. Our structural results indicate that Riplet forms a flexible dimer capable of bridging multiple RIG-I molecules that are bound at variable positions along an RNA duplex, and this may expand the functionality of Riplet under different types of conditions. Indeed, the binding data show that such double-anchoring of Riplet to a single RNA/RIG-I complex can enhance Riplet affinity, which may increase the probability of ubiquitination and facilitate the signaling cascade in certain cellular contexts. On the other hand, our findings clearly show that RIG-I signaling is not dependent on the formation of Riplet-mediated bridges between different RNA molecules, or multiple RIG-I-coated filaments, nor is signaling contingent on Riplet gathering of tangled, interconnected species. Rather, Riplet potentiates RIG-I signaling upon binding to a single RNA, whether bound at one site or two along the RNA lattice.

By examining the structural features of individual Riplet domains and exploring their effects through domain deletions, we provide insights into the function and specificity of each domain. The PrySpry

domain and its interface with RIG-I are clearly visualized in our cryo-EM structures. Consistent with earlier results using a PrySpry fusion protein, a conserved network of aromatic and charged amino acids line the interface between Hel2 helix and the loops projecting from the PrySpry beta sheet. As noted previously, this interface of Riplet is architecturally similar to that formed by the Trim25 E3 ligase. However, the specific amino acids constituting the Trim25 binding surface are different from those mediating Riplet PrySpry interaction with RIG-I. Consistent with this, Trim25 is not observed to ubiquitinate RIG-I and EMSA studies show that Trim25 does not have affinity for RIG-I[23]. Given the differences in amino acid identity and conservation along the PrySpry binding surface, the interface visualized in this study explains the specificity of RIG-I for Riplet rather than the structurally related Trim25 ligase.

Dimerization of the RING motifs results in formation of the compact RING domain that catalyzes ubiquitination of the RIG-I CARDs. Unlike the individual PrySpry domains that can operate separately, the interconnected RING dimer moves as a unit that is joined to the dimerized CC region, which is connected to the individual PrySpry domains by long, flexible tethers (Fig. 4a). Here we show that E3 ligase activity is not the only function of the RING domain, as the domain itself slightly enhances the affinity of Riplet to RIG-I, presumably via a weak interaction with the CARD. Importantly, there are multiple potential sites for Ub conjugation on RIG-I CARDs[12,14,29], and these appear to be redundant, as no individual Lys residue has ever been reproducibly identified as a critical residue for RIG-I activation by undergoing K63-Ub conjugation[8,12,29–32]. Indeed, single mutation of each of CARD Lys residue only slightly reduces the IFN response in the presence of p3SLR14 (Supplementary Fig. 9). This confirms the presence of multiple redundant sites for RIG-I ubiquitination, which may serve to increase the probability that Riplet conjugates Ub to at least one Lys on viral RNA-bound RIG-I CARDs, and ensure an effective IFN anti-viral response.

The CC domain is a conserved component of the Riplet dimer assembly, and yet its deletion is not strictly required for IFN induction in a cell-based IFN reporter assay, which contrasts starkly with the critical role for CC in Trim25[27,28]. This suggests that RING dimerization may be sufficient for stabilizing the Riplet complex, and this may be enhanced via additional interactions between RING and RIG-I CARDs. That said, it is likely that the CC domain provides additional stabilization to the dimerized Riplet assembly, which may help to ensure efficient E3 ligase activity in diverse cellular settings. Intriguingly, the Trim25 RING-CC crystal structure shows RING interactions with the CC domain in apo-Trim-25, suggesting an alternative function of CC other than dimerization for that ligase. It is notable that the Riplet CC domain is shorter and more flexible than the Trim25 CC domain, suggesting that it does not interact with the RING domain, thereby further differentiating the functions of these two related ligases.

Taken together, our mutational functional data and structural analysis suggest a unified model for the role of Riplet in mediating signaling by the RIG-I receptor (Fig. 5b). Upon viral RNA recognition, RIG-I ejects its CARDs, which are then maintained in a sustained "CARDs-out conformation", as described in previous work[21]. Because Riplet is bound directly to RIG-I via the PrySpry domain, it is poised to ubiquitinate the exposed CARDs in a process that does not require RIG-I oligomerization on RNA, nor bridging between RNAs. Ubiquitination of the CARDs prevents them from re-binding to the Hel2i surface on RIG-I and adopting the autoinhibited conformation, thereby driving the pathway forward. In addition, ubiquitination promotes the oligomerization of CARDS from multiple activated RIG-I molecules, enabling them to interact with high affinity to MAVS CARDS, and thereby initiating the signaling relay. This pathway represents a paradigm by which post-translational modifications facilitate signaling, and drive it forward in a directional relay.

## Methods

### Cell lines and plasmids
Parental HEK293T (ATCC, CRL-3216 ™) and Riplet-KO HEK293T cells were grown in high glucose Dulbecco's Modified Eagle Medium (DMEM) supplemented with 10% heat inactivated fetal bovine serum (HI-FBS) at 37 °C with 5% $CO_2$.

The pUNO1 (Invivogen) and Champion™ pET SUMO (Thermo) vectors were used for protein expression in mammalian and *E.coli* cells, respectively. RIG-I and Riplet were cloned into both plasmids. RIG-I and Riplet truncates and mutants were generated using Q5 Site-Directed Mutagenesis Kit (NEB). pUNO1-Riplet-ΔRING (94–432)/ ΔPrySpry (1–248)/ RING (1–93)/ PrySpry (249–432)/ CCto52aaGS (120–183 to GS linker containing 52 amino acids)/ CCto14aaGS (120–183 to GS linker containing 14 amino acids) were cloned. pET-SUMO-Riplet-ΔRING(94–432)/ PrySpry(249–432) were cloned.

### IFN-β induction assay
The IFN-β induction assay was implemented to test RIG-I and Riplet mutants using different RNA duplexes[21]. In brief, 100 μl of HEK293T cells at a concentration of 200,000 cells/ml in Dulbecco's Modified Eagle Medium (DMEM, ThermoFisher) supplemented with 10% heat-inactivated Fetal Bovine Serum (HI-FBS, ThermoFisher) was seeded into each well of 96-well plate (Corning). 24 h after the seeding, the cells of each well were transfected with 3.4 ng of pUNO1-RIG-I/ Riplet unless specified, 3.4 ng of pRL-TK (Renilla luciferase reporter plasmid) and 34 ng of IFN-β/Firefly using the lipofectamine 2000 transfection reagent (ThermoFisher). The RIG-I expression was allowed to proceed for 6 h, at which point the cells of each well were challenged with 0.2 μg of RNAs using lipo2000 reagent. In all, 12–16 h after the stimulation, the HEK293T cells were lysed and the IFN-β induction was measured using the Dual-Luciferase Reporter Assay System (Promega) and a Synergy Neo2 Hybrid Multi-Mode Reader with Gen5 software (Biotek). The IFN-β induction level, the relative luminescence unit (RLU, Fluc/Rluc), is the firefly luciferase activity normalized to the renilla luciferase activity. The data were further processed with GraphPad Prism.

### Riplet-KO cell line generation
Riplet was knocked out in HEK293T cells (ATCC) using conventional CRISPR-Cas9 techniques[33]. The previously reported sgRNA was cloned to PX459 (addgene) and it was expected to cause single indel in exon 1 of Riplet gene located on Chr 17[12]. After puromycin selection, the surviving cells were plated as single cells to 96-well plates. Then the single-cell colonies were applied to IFN-β induction assay and the cells that were not stimulated by p3SLR14 were selected for subsequent evaluation using genotyping and immunoblotting. As there are three copies of Chr 17 in HEK293T cell, the genotyping revealed three different frameshift mutants in the Riplet-KO cells. Consistently, the immunoblotting showed a clear band of Riplet in the lane of parental cells, but no corresponding band in the Riplet-KO cells with the previously reported anti-Riplet primary antibody (Anti-Riplet, Sigma, HPA021576, polyclonal, 1:1000; Anti-GAPDH, Santa Cruz Biotech, sc-47724, monoclonal, 1:1000; Fig. 1c and Supplementary Fig. 2)[13]. All these above suggest the Riplet-KO cells were generated successfully and the cells were used in the IFN-β induction assay.

### Cloning, expression, and purification of RIG-I and Riplet
The human RIG-I and Riplet proteins were expressed and purified[21,22]. In brief, RIG-I and Riplet were fused to an N-terminal 6xHis tag and a SUMO tag, followed by ULP1 digestion site in Champion™ pET SUMO vector, respectively (ThermoFisher Scientific). These constructs were overexpressed in *E.coli* Rosetta™ 2(DE3) Singles™ Competent Cells (Millipore Sigma). RIG-I and Riplet expression was induced by IPTG (0.5 mM) when OD600 reached 0.6 and proceeded for 20–24 h at 16 °C. The pellets were lysed in buffer (25 mM HEPES, pH 8.0, 300 mM

NaCl, 10% Glycerol, 5 mM BME) supplemented with EDTA-free Protease Inhibitor Cocktail (Sigma), followed by nickel affinity chromatography using Ni-NTA Superflow beads (Qiagen). RIG-I and Riplet were treated by ULP1 to remove the SUMO tag. RIG-I was further purified by cation exchange and size exclusion chromatography, using a HiTrap Heparin HP column (GE Healthcare) and then a Superdex 200 Increase 10/300 GL column (GE Healthcare), while Riplet was only applied to size exclusion chromatography using the same column. RIG-I and Riplet were pooled in a storage buffer (25 mM HEPES, pH 7.4, 200 mM NaCl, 5% Glycerol, 5 mM BME) for use in further experiments. RIG-I and Riplet used in cryo-EM studies were pooled in buffer without glycerol.

## RNA preparation

RNA oligonucleotides (p3SLR14; p2SLR14; p3dsRNA24a, p3dsRNA24b) were synthesized in-house using an automated MerMade synthesizer (BioAutomation, Irving, TX, United States) with phosphoramidites from Glen Research using standard phosphoramidite chemistry. Oligonucleotides were deprotected and gel purified as previously described[34], and evaluated for purity by mass spectrometry (Novatia). Briefly, base deprotection was performed in a 1:1 mixture of 30% ammonium hydroxide (JT Baker) and 40% methylamine (Sigma) at 65 °C for 10 min. The supernatant was cooled on ice and evaporated to dryness in a new vial. With the addition of 500 μl absolute ethanol, the solution was evaporated to dryness. In order to deprotect the 2′-OH groups, the pellet was incubated with 500 μl of 1 M solution of tetrabutylammonium fluoride (TBAF) in Tetrahydrofuran (Sigma) at RT for 36 h. Then with addition of 500 μl 2 M sodium acetate (pH 6.0), the solution was evaporated to about 500 μl, extracted with 3 × 800 μl ethyl acetate, following ethanol precipitation. The RNA oligonucleotides were purified using 16% urea-denaturing polyacrylamide gel and its purity was assessed by mass spectrometry (Novatia). The stem-loop RNA (p3SLR50) was in-vitro transcribed, purified, and evaluated[35,36]. In brief, the p3SLR50 was in-vitro transcribed using T7 RNA polymerase with synthetic dsDNA template (Integrated DNA Technologies) containing 2′-OMe modifications on the first two nucleotides of the 5′ terminus of the negative-sense strand. A 100 μl transcription solution contains 1 μg of annealed template, 40 mM Tris-HCl (pH 8.0), 22 mM MgCl2, 10 mM DTT, 2 mM spermidine, 0.01% Triton X-100, 5 mM of each NTPs, 40 U of RNase-OUT™ Recombinant Ribonuclease Inhibitor (Thermo Fisher), and 5 μl of T7 RNA polymerase. With 12-hour incubation at 37 °C, the transcribed p3SLR50 was purified by gel extraction from 12 to 20% urea denaturing polyacrylamide gel and its purity was assessed by mass spectrometry (Novatia). RNA duplexes (p3dsRNA24) and stem-loop RNA (p3SLR14, p2SLR14, and p3SLR50) were annealed and stabilized before use in experiments. Specifically, two-stranded RNA duplexes (260 μM) were annealed by rapidly heating to 99 °C and slowly cooling over 1 hour to 4 °C in annealing buffer (200 mM NaCl) on a Thermocycler. The p3SLR14, p2SLR14, and p3SLR50 were heated to 90 °C for 2 min and then snap-cooled on ice for 30 min. The purity of annealed duplex RNAs was assessed by running samples on a 15% native polyacrylamide gel and visualized by Amersham Typhoon (GE). Sequences of the RNAs used in this study are shown in Supplementary Fig. 1 and Supplementary Table 1.

## Preparation of biotinylated RIG-I

RIG-I was biotinylated for measuring binding affinity to Riplet. With Avi tag (N- GLNDIFEAQKIEWHE-C) fused to the N-terminal of RIG-I, the purification of Avi-tagged RIG-I was described above. The Avi tag was biotinylated by BirA in buffer (40 μM Avi-RIG-I, 5 μg BirA, 50 mM bicine, pH 8.3, 10 mM ATP, 10 mM MgOAc, 50 μM d-biotin) at 30 °C for 40 min (Avidity). The biotinylated RIG-I was further purified by size exclusion chromatography, using a Superdex 200 Increase 10/300 GL column (GE Healthcare), and pooled in a storage buffer (25 mM HEPES, pH 7.4, 200 mM NaCl, 5% Glycerol, 5 mM BME) for use in further experiments. The biotinylated RIG-I was evaluated by immunoblotting.

## Surface plasmon resonance

The interactions between Riplet and RIG-I:p3SLR14, RIG-I:p3dsRNA24, and apo-RIG-I were characterized by SPR using Pioneer FE SPR System with Pioneer FE software (Sartorius). The biotinylated RIG-I was incubated with p3SLR14 and p3dsRNA24 in ratio of 1:1.5 and 2.4:1 at 4 °C overnight, respectively, following 500 nM samples were immobilized to SADH sensor chip already immobilized with Streptavidin (Sartorius). The dilutions (10, 5, 2, 1, 0.5, 0.25, 0.125 μM in HBS-EP+, cytiva) of Riplet-FL, Riplet-ΔRING, and Riplet-PrySpry were applied to the chip at flow rate of 50 μl/min (279 μl) and dissociation time of 180 s in OneStep mode, which allows the measurement of binding affinity from a single analyte concentration injection. Data analysis was performed using Qdat software (Sartorius). The kinetics including rate constants of association ($K_a$) and dissociation ($K_d$), and binding affinity ($K_D = K_d/K_a$) were obtained by fitting 1:1 model, which were summarized in Supplementary Fig. 3 and Supplementary Table 2.

## Electrophoretic mobility shift assay

The EMSA was implemented to test the formation of RIG-I:RNA:Riplet complex presence of different RNAs. 2.5 μM RIG-I and 50 ng RNA (p2SLR14, ~450 nM; p3dsRNA24, ~300 nM) were incubated at room temperature for 30 min, then 2.5 uM Riplet was added to the solution following incubation at RT for 30 min. Formation of the complex was assessed by running samples on a 6% native polyacrylamide gel staining with GelRed (Biotium) and visualized by Amersham Typhoon (GE).

## Purification of RIG-I:RNA:Riplet complexes

To prepare RIG-I:RNA:Riplet complexes, purified RIG-I, RNA, and Riplet were mixed in an 8:1:8 molar ratio and incubated overnight at 4 °C. The RIG-I:RNA:Riplet complexes were separated using a Superdex 200 Increase 10/300 GL column (GE Healthcare). The complexes were pooled in size-exclusion buffer (25 mM HEPES, pH 7.4, 200 mM NaCl, 5 mM BME) and concentrated with a 50 kD Amicon Ultra-0.5 Centrifugal Filter Unit (Millipore Sigma). The absorbance was estimated using a Nanodrop (ThermoFisher) for concentration quantification.

## Cryo-EM sample preparation

To prepare for freezing cryo-EM grids, the fresh samples were obtained and concentrated. The RIG-I:RNA:Riplet complexes were concentrated to 1.1 mg/ml. The Quantifoil holey carbon R1.2/1.3 300 mesh Cu grids (Ted Pella) were glow discharged using the PELCO easiGlow™ Glow Discharge Cleaning System (Ted Pella) for 35 s at 25 mA. With purified RIG-I:RNA:Riplet complexes (3.5 μl) applied onto the grids, the grids were blotted and plunged to the liquid ethane for flash freezing using a Vitrobot Mark IV (ThermoFisher). The blotting conditions for all grids were similar, under conditions of 22 °C and 100% humidity with force −4. Blotting times were 2 s for RIG-I:RNA:Riplet complexes. The frozen grids were all transferred and kept in liquid nitrogen prior to use in data collection.

## Cryo-EM data acquisition

Cryo-EM data were acquired at the HHMI Janelia Research Campus on a Titan Krios transmission electron microscope (ThermoFisher) operating at 300 keV and equipped with a Gatan K3 Summit direct electron detector using SerialEM software at super-resolution mode[37]. For the RIG-I:RNA:Riplet complexes, 3420 micrographs were collected at a nominal magnification of 105,000 ×, corresponding to calibrated pixel size of 0.839 Å/pix, with a defocus range of −1.0 to −2.5 μm. Each micrograph contains 40 frames and was collected with an exposure rate of 8.51 e⁻/pix/s and total electron exposure of 60 e⁻/Å². The statistics of data acquisition are summarized in Supplementary Table 3.

 

## Cryo-EM data processing

The dataset was processed through Relion[38]. The micrographs were dose-weighted and beam-induced motion corrected through MotionCor2[39]. The non-dose-weighted and motion corrected micrographs were used to estimate the CTF parameters using CTFFIND4[40]. Micrographs were selected based on the Total motion and image resolution, and further selected by manual screening.

For the RIG-I:RNA:Riplet complex, 1,852,320 particles were initially picked from 3330 selected micrographs using the 3D reference of RIG-I:RNA:Riplet complex reconstructed from a cryo-EM dataset collected on a 200 KeV Glacios with Gatan K2 Summit direct electron detector located at Yale Cryo-EM facility. The particles were subjected to several rounds of 2D classification. Particles from all 2D classes containing a 2:1:2 RIG-I:RNA:Riplet were selected. The selected particles were then used to generate the initial model through Relion, and subjected to several rounds of 3D classification. Half of map is at high resolution, while the other half show low-resolution feathers. To improve the resolution, the worse half in the 3D map generated above was kept through Chimera[41], and a soft mask was applied to implement the 3D classification without particle alignment. Three different maps were generated. The end-end complex, end-transition complex and end-inter complex contained 204,993, 68,414, and 19,143 particles, which were subjected to 3D refinement, Ctfrefine, and Byaesian polishing, following a final 3D refinement[38,42]. The postprocessing yielded a map at global resolution of 3.2 Å, 4.0 Å, and 3.9 Å, sharpened with B-factor of −76.00 Å², −131.39 Å² and −88.73 Å², according to the FSC = 0.143 criterion[43]. The strategy and statistics of data processing are summarized in Supplementary Fig. 4 and Supplementary Table 3.

## Model building, refinement, and validation

For the end-end complex, PDB ID 7TNX and 7JL1 were used as the initial search model for building into the cryo-EM map, using molrep in the ccpem software suit, respectively[21,22,44]. The end-end complex was used as the initial search model for end-semi-closed-end complex and end-inter complex. All models were built and manually adjusted in Coot[45]. Then the models were refined against the cryo-EM maps using phenix.real_space_refine within phenix and refmac5 within the ccpem software suit[46]. Models were validated using Comprehensive validation (cryo-EM) in phenix[46,47]. All maps and models were further validated through the PDB validation server. The statistics of model building, refinement, and validation are summarized in Supplementary Table 3. All the figures were generated using PyMOL (http://www.pymol.org/) and Chimera[41].

## Sequence alignment

The Riplet protein sequences from multiple mammals were aligned using Clustal Omega[48], and the results were rendered using ESPript 3.0 (https://espript.ibcp.fr)[49].

## Structure prediction by ColabFold

The tertiary structures of full-length Riplet (1–432), RING domain (1–94) and CC domain (120–240) were predicted using ColabFold: AlphaFold2 using MMseqs2 (https://colab.research.google.com/github/sokrypton/ColabFold/blob/main/AlphaFold2.ipynb)[25]. To ensure the predication of dimer, the protein sequence was added twice, separated by colon symbol. The structures with highest pIDDT values were selected for further analysis.

## Reporting summary

Further information on research design is available in the Nature Portfolio Reporting Summary linked to this article.

## Data availability

The atomic coordinates and cryo-EM maps generated in this study have been deposited in EMDB and PDB as follows: end-end RIG-I:p3dsRNA24:Riplet complex (EMDB: 29823, PDB: 8G7T [https://doi.org/10.2210/pdb8G7T/pdb]), end-semi-closed-end RIG-I:p3dsRNA24:Riplet complex (EMDB: 29824, PDB: 8G7U [https://doi.org/10.2210/pdb8G7U/pdb]) and end-inter RIG-I:p3dsRNA24:Riplet complex (EMDB: 29825 [https://www.ebi.ac.uk/emdb/EMD-29823], PDB: 8G7V [https://doi.org/10.2210/pdb8G7V/pdb]). The raw cryo-EM micrographs have been deposited in EMPIAR (EMPIAR-11494). The raw micrographs of EMSA and immunoblotting have been provided as source data. The raw data of IFN reporter assay have been provided as source data. Source data are provided with this paper.

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

## Acknowledgements

We thank O. Fedorova and S. Fergione for synthesizing the RNA oligonucleotides used in the study. We thank Ling Xu for providing p3SLR50. We thank O. Fedorova for helping conduct the SPR binding assay. We thank S. Yang and X. Zhao at the HHMI Janelia CryoEM Facility for their help in microscope operation and data collection. We thank Chengxin Zhang for helping with model prediction. We thank Chunxiang Wu and Prof. Yong Xiong for SEC-MALS experiment. We thank Ananth Kumar for helping with protein preparation. B.G. was generously supported by the Deutsche Forschungsgemeinschaft (DFG, German Research Foundation). This work was supported by HHMI and by NIH Grant R01AI131518.

## Author contributions

A.M.P. and W.W. designed the experiments. W.W. performed the experiments and analyzed the data. B.G. led the experiments for paper revision. B.G. performed the IFN reporter assay and analyzed the data. R.G. expressed recombinant mutant Riplet constructs. B.G. and R.G. performed the SEC-MALS assay and analyzed the data. A.M.P., W.W., and B.G. wrote the paper.

## Competing interests

A.M.P. has founded a company (RIGImmune) to develop SLRs as therapeutic agents. Yale University and A.M.P. have an issued patent on p3SLR14 (17/709,282), which was used in this study. The remaining authors declare no competing interests.
