## [Peer Review File · Nature Communications]

The E3 ligase Riplet promotes RIG-I signaling independent of RIG-I oligomerizationREVIEWER COMMENTS

Reviewer #1 (Remarks to the Author):

The authors used biochemistry and structural biology to study how RIPLET binds and regulates RIGI. They first found the activation of RIGI by RIPLET is comparable upon treatment with either short dsRNA or with long dsRNA, indicating RIPLET works on non-filamentous RIGI. Then they tested the individual domains of Riplet on RIG-I activation in the presence of short dsRNA and found RING and PrySpry domains are critical while CC is dispensable. Next they measured the binding affinity of different truncations of RIPLET with RIG-I:p3SLR14(1 RIG-I recruitment) RIG-I:p3SLR24(2 RIG-I recruitment), or Apo-RIG-I, and found the affinity is comparable between RIG-I:p3SLR14 and the Apo-RIG-I, indicates RIPLET may bind and work on apo RIG-I. They then solved the cryo-EM structures of RIPLET with RIG-I/RNA complex, and combined AlphaFold prediction to propose a working model of RIG-I regulation by RIPLET.

This is an interesting work and provides some novel insights, however, major issues must be addressed prior acceptance to ensure the conclusion is well supported by experimental data.

1. The authors conclude RIPLET form a stable complex with app RIGI given the binding affinity is comparable with RIG-I:p3SLR14 complex, however, direct evidence is needed to define the formation of "a stable complex". For example, could they observe the two proteins interact with pull-down, or native page assays? Is the binding affinity mediated by PrySpry-Hel2 interaction? If they mutate the residues at the interface, will the binding affinity significantly reduced?
2. Because the RING domain deletion slightly reduced binding affinity, the authors propose it enhances Riplet-RIG-I interaction through binding to CARD domain. But how this explain the difference between RipletRING-RIG-I:p3SLR14 (1.2uM) and Riplet-FL-Apo-RIG-I (0.7uM), given the CARD of apoRIGI is not accessible to RING? Is it possible that the RING domain dimerization itself contributes to the Riplet-RIG-I interaction?
3. The authors made a construct to replace CC domain with a GS linker, which has no effect on IFN response. But CC domain deletion reduces the binding affinity. To better understand the importance CC domain, could the author measure the binding affinity of the GS linker construct with short and long dsRNAs? This should be different from the Riplet-Pspry when both CC and RING are removed.
4. Does the GS linker protein also forms a dimer on SEC or MALS? this is related with question #2 to test whether the RING domain mediate Riplet dimerization, and to test the ColabFold predicted model of dimerization by RING domain.
5. The ColabFold model needs more experimental test. If the authors mutate the predicted dimerization interface, could they observe proteins form monomer rather than dimer, on SEC/MALS?
6. The author stated the interaction is not surprising given the interface is accessible in apo-RIG-I. Given the weak affinity, it should be discussed that the cellular concentration of the two proteins matter in terms of complex formation. Is the basal expression level well-known? How they differ in different cell types and infection conditions?
7. The discussion part is too long.

Reviewer #2 (Remarks to the Author):

The manuscript by Wang et al, investigates anew how E3 ligase RIPLET (which ubiquitinates the RIG-I CARDS) interacts with RIG-I and what association of the two molecules is necessary and sufficient for productive signalling and interferon induction.

A key observation of the paper is that the degree of IFN- β induction in HEK293T cells is essentially the same whether stimulation is with p3SLR14 (which can only end-bind one RIG-I) or p3dsRNA24 (which can bind two RIG-I molecules)(Supp. Fig. 1b). The authors conclude that it is sufficient that one monomer of the homodimeric RIPLET binds a single RNA bound RIG-I to induce signalling, despite the surprising fact that this 2:1 interaction is 100 x weaker than the 2:2 binding of RIPLET to p3dsRNA24 bound to two RIG-I molecules. Furthermore, the homodimeric coiled-coil of RIPLET is dispensable. This is in contrast to previous work by the Wu group (Cadena et al, Kato et al) that RIPLET only works when bridging RIG-I molecules that coat long dsRNA as a filament. Finding that the weak p3SLR14-RIG-I-RIPLET 1:2 complex is too unstable for structure determination, the authors obtain three distinct cryo-EM structures of the more stable 2:2 complex. All complexes have one full end-bound RIG-I but differ in the way the second RIG-I is bound to the other end of the RNA (full end-bound, semi-closed end-bound, internal). The only part of RIPLET seen is the PrySpry domain, which always binds in the same way to the Hel2 domain of each RIG-I (and similar to that observed previously by Kato et al), but the two PrySpry domains differ in their relative orientation in the three complexes. Mutations in the interface, underscore its importance for IFN activation. Interestingly, the authors observe that RIPLET can also bind to apo-RIG-I using the same interface. Finally, using AlphaFold modelling of the unobserved parts of RIPLET (the homodimeric coiled-coil and RING domains) the authors propose a model in which inter-domain flexibility of the three regions of RIPLET enable it to adapt to bind two RIG-I molecules in different configurations and ubiquitinate the CARDS, even though binding to only one RIG-I is sufficient for signalling.

The most significant and novel contribution of this work is to show that homodimeric RIPLET mediates downstream signalling of a single RIG-I that is bound to a short target RNA and that this is not dependent on the CC domain for RIPLET dimerisation. The fact that the RIPLET PRYSPRY interface is essentially the same as previously described and that the RIPLET homodimer forms a tighter complex by bridging neighbouring RIG-I molecules (implied by the previous work) detracts somewhat from the novelty of the structural work.

(1) The authors clearly show, in line with their previous work, that short ppp-dsRNA that only bind one RIG-I can activate interferon and that this is mediated by binding of only one PRYSPRY of the RIPLET homodimer. However, this is a very low affinity interaction. Furthermore, activation of MAVS requires at least four such ubiquitinated complexes to come together in a tetramer. On the other hand, a target RNA that can bind 2 or even 4 RIG-I molecules (end bound or internally bound) has a much higher affinity for the RIPLET homodimer (and hence more readily gives a structure, as here or in Kato et al) and fewer such molecules are required to activate MAVS. It is a bit surprising that the results of this paper, suggest that both RNAs are equally good in inducing interferon. One might think that dependent on the length of target RNA, there would be a different concentration dependence of interferon induction and that longer RNAs would be more effective at limiting RNA concentrations. Considerations of this kind seem to be missing from the discussion and the interferon induction experiments are done with a fixed concentration of RNA. Perhaps this could be thought through more and it may lead to some reconciliation between the results presented by this group and those from the Wu group.

(2) Kato et al reported a previous structure of RIPLET PRYSPRY domain bound to RIG-I itself internally bound on dsRNA. Although the current work uses full-length RIPLET and RIG-I and the previous work used a RIG-I fused to RIPLET PRYSPRY domain, the end structures appear to be basically the same regarding the interaction. Indeed, it is said that the two structures are 'consistent' but a direct comparison should be made and also an acknowledgement of the priority of the Kato structure.

(3) The data in Supp Figure 1b is key to the argument and should be moved into Figure 1.

(4) On initial reading of the main text it is unclear exactly how the CC-removed mutants have been constructed. Only in the Methods do we learn that Ccto52aaGS means 120-183 deleted and replaced by (GS)₂₆ (and similar for Ccto14aaGS). This should be fully explained in the main text. (NB there is a typo in the Methods concerning Ccto14aaGS, where it wrongly says the GS linker is 52 aa). Perhaps more importantly biochemical or biophysical data (e.g. Sec-MALLS) should be presented showing these constructs are still homodimeric by virtue of the RING domains, perhaps similar to Figure 1c.

(5) The authors at several points suggest that RIPLET and TRIM25 are 'similar'. Whereas the linear architecture might suggest this, the 3D architecture shows some significant difference, some of which are mentioned. Perhaps it can also be highlighted that RIPLET has a much shorter coiled-coil region that does appear to form a platform for binding of the RING or PRYSPRY domains and that the CC is predicted to be parallel rather than antiparallel as in TRIM25.

(6) It is now accepted practice to include PAE plots when reporting structures of complexes predicted by AlphaFold to indicate whether inter-domain interactions are predicted with high confidence. This should be added to Figure 4, in particular with respect to the RING domain and CC dimers.

(7) The title of Supplementary Fig. 5: Unique features of the Riplet CC domain, does not seem appropriate to the figure itself. Please check.

(8) Sometimes it is not rationalised why a pp-dsRNA is used rather than a ppp-dsRNA.

Reviewer #3 (Remarks to the Author):

Wang et al demonstrated in this manuscript that Riplet promotes RIG-I activation through a mechanism that does not involve RIG-I filament formation. However, given the extensive research on RIG-I and Riplet, this finding may not be considered significant. Additionally, there are several concerns regarding the data presented.

1. The authors argue that Riplet can activate RIG-I without the presence of a filament, as supported by their discovery that a 14-bp stem-loop RNA can induce IFN expression that relies on Riplet. However, to verify their claims, the authors should conduct quantitative biochemical and cellular RIG-I signaling assays using RNA substrates of different lengths. The efficiency of RIG-I activation in the absence and presence of a filament must be compared. After all, strong claims require strong evidence.

2. The structures of RIG-I:RNA:Riplet resolved in this paper exhibit considerable resemblance to those obtained in a recent study (Kato et al., PMID: 33373584), which employed artificially fused proteins. Especially, the interface and critical interacting residues revealed by Wang et al. are nearly identical to those previously reported. Thus, the new insights provided by this work are limited to the validation of natural interface between RIG-I and Riplet.

3. Figures 1a and 1b demonstrate that Riplet expression leads to a more than ten-fold increase in IFN induction in the absence of p3SLR14 stimulation, indicating a short RNA-independent effect of Riplet. However, in the presence of p3SLR14 stimulation, Riplet expression results in less than a five-fold increase of IFN induction. These results suggest that Riplet may not have a significant role in p3SLR14-induced IFN expression.

Response to the Reviewer Comments.

We thank all the reviewers for their careful analysis of the manuscript and for their many helpful suggestions. Below we provide our enumerated responses to each of the individual reviewer comments (Our responses are in blue font).

Reviewer #1:

1. The authors conclude RIPLET form a stable complex with app RIGI given the binding affinity is comparable with RIG-I:p3SLR14 complex, however, direct evidence is needed to define the formation of “a stable complex”. For example, could they observe the two proteins interact with pull-down, or native page assays? Is the binding affinity mediated by PrySpry-Hel2 interaction? If they mutate the residues at the interface, will the binding affinity significantly reduced?

Perhaps we were unclear in our description of the binding assays that were conducted to examine the complexes. As now stated clearly in the text, we have provided direct evidence for a complex between RIG-I and Riplet, in the form of quantitative SPR assays, which is why we report a value for the K_d . Pull-down experiments would not provide this level of precision. Given the observed moderate affinity, however ($\sim 1 \mu\text{M}$ in the case of Riplet and apo-RIG-I), it would be difficult to accurately interpret effects of mutations because micromolar affinity is already at the border of detectable binding by EMSA. Furthermore, the cryo-EM structures (both in this work, and in Kato, see Reviewer 2 comment 2 below) both establish formation of a specific PrySpry-Hel2 interface. It is already known that PrySpry binding does not induce conformational change of Hel2 in RIG-I (see Figure at right), suggesting that the same PrySpry-Hel2 interaction mediates Riplet binding to apo-RIG-I. That said, we agree that more care should be taken in making statements about the strength of RIPLET-RIG-I complex affinity. For this reason, we now say that “full-length Riplet recognizes both apo-RIG-I and RIG-I:p3SLR14 complex with similar, moderate affinity” (page 6) and “our electrophoretic mobility shift assay (EMSA) indicated that this complex was just below the level of stability needed to obtain a complex under cryo-EM conditions” (page 7). We removed potentially misleading descriptions of complex stability from Discussion section (page 11).

Hel2 and PrySpry of RIG-I:p3dsRNA24:Riplet structure
Hel2 of duck apo-RIG-I, PDB code: 4a2w

Graphic for Rev1, point 1. Superimposed Hel2 domains between apo-RIG-I and RIG-I from RIG-I:RNA:Riplet complex

Graphic for Rev1, point 2: Apo-RIG-I structure (PDB: 4a2w)

2. Because the RING domain deletion slightly reduced binding affinity, the authors propose it enhances Riplet-RIG-I interaction through binding to CARD domain. But how this explain the difference between RipletDRING-RIG-I:p3SLR14 ($1.2\mu\text{M}$) and Riplet-FL-Apo-RIG-I ($0.7\mu\text{M}$), given the CARD of apoRIGI is not accessible to RING? Is it possible that the RING domain dimerization itself contributes to the Riplet-RIG-I interaction?

Here we have been relatively conservative in our interpretations, noting that RING domain binding to its substrate (CARDs) might be expected to slightly enhance its affinity, as supported by the SPR data showing that RING deletion decreases Riplet affinity from 0.7 to $1.2 \mu\text{M}$. The reviewer asks why the affinities are not identical (1.2 vs $0.7 \mu\text{M}$), presumably based on the assumption that in apo-RIG-I the CARDs are buried within the autoinhibited structure and not accessible by the

RING domain. However, this assumption is not correct. Even when the CARDs are engaged with Hel2i in apo-RIG-I, the RING domain binding site remains accessible (see Figure at left).

3. The authors made a construct to replace CC domain with a GS linker, which has no effect on IFN response. But CC domain deletion reduces the binding affinity. To better understand the importance CC domain, could the author measure the binding affinity of the GS linker construct with short and long dsRNAs? This should be different from the Riplet-Pspry when both CC and RING are removed.

While this is an interesting question, dissecting the specific role of the CC domain on RNA binding specificity is outside the scope of this study. As additional structural information becomes available on the full-length protein, it will be useful to revisit this idea.

4. Does the GS linker protein also forms a dimer on SEC or MALS? this is related with question #2 to test whether the RING domain mediate Riplet dimerization, and to test the ColabFold predicted model of dimerization by RING domain.

In response to this comment, we tested oligomerization of the Ccto14aaGS mutant using SEC-MALS, finding that it elutes as a monomer from the SEC column (data not shown). This was in fact expected because the RING domains of Riplet homologues (Trim5 α and Trim25) were also found to be monomeric in solution using SEC-MALS, as the method often fails to capture complexes in the micromolar affinity range^{1,2}. However, Trim5 α and Trim25 RING domains crystallize as dimers and Trim25 RING dimerization has been carefully evaluated by NMR. In the latter case, dimerization of RING domain was shown to be indispensable for Ub conjugation^{1,2}. Given that the homologous Riplet RING was predicted to be a dimer sharing the same architecture as TRIM ligases (Supp Fig 7), and the Ccto14aaGS mutant can still signal when RIG-I is bound to p3SLR14 (Main Fig 1e, 1f, 1g), it's a reasonable assumption that a Riplet RING dimer can form and might be observable under certain conditions (crystal, high concentration). By extension to Trim25, it is also reasonable to suggest that the Riplet RING dimer is important for Ub conjugation. For clarity on these issues, we have changed the corresponding descriptions in the text (page 10).

5. The ColabFold model needs more experimental test. If the authors mutate the predicted dimerization interface, could they observe proteins form monomer rather than dimer, on SEC/MALS?

We would contend that the predicted models of dimeric RING and CC domains are reasonably convincing in terms of standard metrics, such as the the high pLDDT, low PAE values and high conservation of residues in the monomer-monomer interface (Main Fig 4b, Supp Fig 8). In addition, the conserved Riplet RING domain appears to dimerize in the same way as other TRIM E3 ligases that have been structurally characterized in great detail (Supp Fig 7). Therefore, it's a reasonable assumption that mutations in the predicted RING monomer-monomer interface would disrupt the dimerization of the RING domain. The Riplet CC domain is predicted to maintain its dimerization by almost 20 hydrophobic residues (Main Fig 4f), which would make meaningful mutations difficult to create. All these residues are highly conserved among species (Supp Fig 8), strengthening the predicted model of CC dimer. Perhaps most relevant to our resubmission is that specific features of the dimerization interface in this model are not central to the mechanistic claims made in our manuscript, so delays incurred by additional structure-function experiments would not seem merited in this particular manuscript.

6. The author stated the interaction is not surprising given the interface is accessible in apo-RIG-I. Given the weak affinity, it should be discussed that the cellular concentration of the two proteins matter in terms of complex formation. Is the basal expression level well-known? How they differ in different cell types and infection conditions?

To our knowledge, RIG-I and Riplet protein concentrations in cells have not been measured before, but the RNA transcript levels were estimated using RNA-seq (Human protein atlas, www.proteinatlas.org/). According to the protein atlas database, RIG-I and Riplet are expressed in most organs, tissues and cell types. On average, the RNA transcript level of Riplet is 3-4 fold more than RIG-I, but how this is reflected in protein expression would be very difficult to determine, and is beyond the scope of this study.

7. The discussion part is too long.

We appreciate the importance of brevity in writing the Discussion, and we therefore drafted this part of the paper so that it conforms to a length that is typical for the journal. For this reason, and to ensure that we provide a lucid description of the working model for Riplet - RIG-I activation, we hesitate to reduce the length of the Discussion.

Reviewer #2:

1. The authors clearly show, in line with their previous work, that short ppp-dsRNA that only bind one RIG-I can activate interferon and that this is mediated by binding of only one PRYSPRY of the RIPLET homodimer. However, this is a very low affinity interaction. Furthermore, activation of MAVS requires at least four such ubiquitinated complexes to come together in a tetramer. On the other hand, a target RNA that can bind 2 or even 4 RIG-I molecules (end bound or internally bound) has a much higher affinity for the RIPLET homodimer (and hence more readily gives a structure, as here or in Kato et al) and fewer such molecules are required to activate MAVS. It is a bit surprising that the results of this paper, suggest that both RNAs are equally good in inducing interferon. One might think that dependent on the length of target RNA, there would be a different concentration dependence of interferon induction and that longer RNAs would be more effective at limiting RNA concentrations. Considerations of this kind seem to be missing from the discussion and the interferon induction experiments are done with a fixed concentration of RNA. Perhaps this could be thought through more and it may lead to some reconciliation between the results presented by this group and those from the Wu group.

We agree with the reviewer that titrating short and long dsRNAs in the IFN reporter assay would strengthen the paper, and to this end we have now compared the stimulatory influence of p3SLR14 to that of longer RNAs p3dsRNA24 and p3SLR50. Indeed, the IFN responses stimulated by RNAs of different length and scaffold design are similar, indicating that short RNAs can induce Riplet-mediated RIG-I responses as well as long RNAs. Please see Main Fig 1a and page 5.

2. Kato et al reported a previous structure of RIPLET PRYSPRY domain bound to RIG-I itself internally bound on dsRNA. Although the current work uses full-length RIPLET and RIG-I and the previous work used a RIG-I fused to RIPLET PRYSPRY domain, the end structures appear to be basically the same regarding the interaction. Indeed, it is said that the two structures are 'consistent' but a direct comparison should be made and also an acknowledgement of the priority of the Kato structure.

In response to this important point, a new supplementary figure has been added to compare the structures. Indeed, the binding interfaces revealed by the two structures are almost identical (Main Supp Fig 6c), reinforcing the idea that Riplet recognizes RIG-I regardless of its RNA-bound status. We also modified the text to acknowledge the priority of Kato's structure (Page 9).

3. The data in Supp Figure 1b is key to the argument and should be moved into Figure 1.

As described above, we have now titrated different RNAs in the IFN reporter assay. As suggested, this new data, which includes the information previously presented in Supp Fig 1b, is now displayed in Main Fig 1a.

4. On initial reading of the main text it is unclear exactly how the CC-removed mutants have been constructed. Only in the Methods do we learn that CCto52aaGS means 120-183 deleted and replaced by (GS)26 (and similar for

CCto14aaGS). This should be fully explained in the main text. (NB there is a typo in the Methods concerning CCto14aaGS, where it wrongly says the GS linker is 52 aa). Perhaps more importantly biochemical or biophysical data (e.g. SEC-MALLS) should be presented showing these constructs are still homodimeric by virtue of the RING domains, perhaps similar to Figure 1c.

We thank the reviewer for these suggestions. We have now added more information about the CC mutant design in the main text (page 6). We also tested the oligomeric state of the GS mutant using SEC-MALS and find that it elutes as monomer from the SEC column, as seen for other E3 ligases in this family (See Reviewer #1, comment #4). As a result, we have changed our description of "RING dimerization independent of the CC domain" (page 10).

5. The authors at several points suggest that RIPLET and TRIM25 are 'similar'. Whereas the linear architecture might suggest this, the 3D architecture shows some significant difference, some of which are mentioned. Perhaps it can also be highlighted that RIPLET has a much shorter coiled-coil region that does appear to form a platform for binding of the RING or PRYSPRY domains and that the CC is predicted to be parallel rather than antiparallel as in TRIM25.

Thanks for the comment. We have added this information to the main text (page 10).

6. It is now accepted practice to include PAE plots when reporting structures of complexes predicted by Alphafold to indicate whether inter-domain interactions are predicted with high confidence. This should be added to Figure 4, in particular with respect to the RING domain and CC dimers.

We have added the PAE plots to Figure 4b.

7. The title of Supplementary Fig. 5: Unique features of the Riplet CC domain, does not seem appropriate to the figure itself. Please check.

We agree and have changed the title from "Unique features of the Riplet CC domain" to "Riplet CC domain accommodates a range of distances between two PrySpry domains".

8. Sometimes it is not rationalised why a pp-dsRNA is used rather than a ppp-dsRNA.

Thank you for this feedback. On re-reading the text we realized that specific description of attempted cryo-EM studies on ppSLR14-RIG-I complexes introduced unnecessary confusion. In fact, we attempted to obtain structures with both di- and tri-phosphorylated SLR14 variants to no avail during this part of the study, so we simply refer to SLR14 complexes generically in the relevant section of the text (page 7). But to answer your question specifically, our previously published work has established that both 5'pp and 5'ppp-dsRNA are almost identical, behaving as potent RIG-I agonists that bind to the receptor in a similar manner³. For example, they both bind to RIG-I with pM affinity, and they induce comparably strong RIG-I-mediated IFN response⁴⁻⁷. We have tried to be more clear about this now in the manuscript.

Reviewer #3:

1. The authors argue that Riplet can activate RIG-I without the presence of a filament, as supported by their discovery that a 14-bp stem-loop RNA can induce IFN expression that relies on Riplet. However, to verify their claims, the authors should conduct quantitative biochemical and cellular RIG-I signaling assays using RNA substrates of different lengths. The efficiency of RIG-I activation in the absence and presence of a filament must be compared. After all, strong claims require strong evidence.

We appreciate these suggestions and agree that comparing short and long RNA ligands in the cell-based IFN assay would strengthen the paper. To this end we have now tested p3SLR14, p3dsRNA24 and long RNA p3SLR50. Please see Reviewer #2, comment #1, as presented in the new Main Fig 1a.

2. The structures of RIG-I:RNA:Riplet resolved in this paper exhibit considerable resemblance to those obtained in a recent study (Kato et al., PMID: 33373584), which employed artificially fused proteins. Especially, the interface and critical interacting residues revealed by Wang et al. are nearly identical to those previously reported. Thus, the new insights provided by this work are limited to the validation of natural interface between RIG-I and Riplet.

We are glad that the reviewer raises this important point. Although the two studies reveal an identical PrySpry-Hel2 interface, it was never obvious that design of the artificial fusion protein enabled PrySpry and Hel2 domains to assemble appropriately and it remained important to evaluate the natural Riplet-RIG-I interface using the two separate proteins. In accomplishing this, we have established structurally that Riplet recognizes end-bound RIG-I in a highly specific manner and we reveal the natural interface without forcing any particular design on the construct. Furthermore, by combining information from Kato's structure with the structural and binding data from this paper, we are able to conclude that Riplet recognizes RIG-I regardless of its RNA-bound state (e.g. apo-RIG-I, end-bound RIG-I, internally-bound RIG-I). Together, these findings provide valuable insights into the molecular mechanism of RIG-I activation by Riplet, and are therefore not limited to a simple validation of a molecular interface.

3. Figures 1a and 1b demonstrate that Riplet expression leads to a more than ten-fold increase in IFN induction in the absence of p3SLR14 stimulation, indicating a short RNA-independent effect of Riplet. However, in the presence of p3SLR14 stimulation, Riplet expression results in less than a five-fold increase of IFN induction. These results suggest that Riplet may not have a significant role in p3SLR14-induced IFN expression.

We respectfully disagree with the reviewer on this point. We have established that Riplet is indeed indispensable for p3SLR14-mediated RIG-I signaling, as a Riplet knockout almost abolishes the response (Main Fig 1b), as observed in other studies^{8,9}. The reviewer questions why massive Riplet overexpression leads to a slightly higher fold change in IFN activation in the absence of p3SLR14 (Main Fig 1a, 1b), and we agree that this is an odd finding. But IFN induction in the absence of RNA has been seen before upon overexpression of pathway components (for example, transfected MAVS overexpression also turns on the response), and so we think it likely that these results may not be physiologically-relevant. This is supported by the fact that endogenous Riplet does not induce signaling at all under physiological conditions in the absence of RNA ligand (new Main fig 1b, 'No Riplet' group).

References

- 1 Koliopoulos, M. G., Esposito, D., Christodoulou, E., Taylor, I. A. & Rittinger, K. Functional role of TRIM E3 ligase oligomerization and regulation of catalytic activity. *EMBO J* **35**, 1204-1218 (2016). <https://doi.org:10.15252/embj.201593741>
- 2 Yudina, Z. *et al.* RING Dimerization Links Higher-Order Assembly of TRIM5alpha to Synthesis of K63-Linked Polyubiquitin. *Cell Rep* **12**, 788-797 (2015). <https://doi.org:10.1016/j.celrep.2015.06.072>
- 3 Goubau, D. *et al.* Antiviral immunity via RIG-I-mediated recognition of RNA bearing 5'-diphosphates. *Nature* **514**, 372-375 (2014). <https://doi.org:10.1038/nature13590>
- 4 Rawling, D. C., Fitzgerald, M. E. & Pyle, A. M. Establishing the role of ATP for the function of the RIG-I innate immune sensor. *Elife* **4** (2015). <https://doi.org:10.7554/eLife.09391>
- 5 Linehan, M. M. *et al.* A minimal RNA ligand for potent RIG-I activation in living mice. *Sci Adv* **4**, e1701854 (2018). <https://doi.org:10.1126/sciadv.1701854>
- 6 Ren, X., Linehan, M. M., Iwasaki, A. & Pyle, A. M. RIG-I Selectively Discriminates against 5'-Monophosphate RNA. *Cell Rep* **26**, 2019-2027 e2014 (2019). <https://doi.org:10.1016/j.celrep.2019.01.107>
- 7 Wang, W. & Pyle, A. M. The RIG-I receptor adopts two different conformations for distinguishing host from viral RNA ligands. *Mol Cell* **82**, 4131-4144 e4136 (2022). <https://doi.org:10.1016/j.molcel.2022.09.029>
- 8 Shi, Y. *et al.* Ube2D3 and Ube2N are essential for RIG-I-mediated MAVS aggregation in antiviral innate immunity. *Nat Commun* **8**, 15138 (2017). <https://doi.org:10.1038/ncomms15138>
- 9 Cadena, C. *et al.* Ubiquitin-Dependent and -Independent Roles of E3 Ligase RIPLET in Innate Immunity. *Cell* **177**, 1187-1200 e1116 (2019). <https://doi.org:10.1016/j.cell.2019.03.017>

REVIEWERS' COMMENTS

Reviewer #1 (Remarks to the Author):

The authors have addressed my concerns.

Reviewer #2 (Remarks to the Author):

The authors have satisfactorily answered the referee's queries, including adding new data on the concentration dependence of different length RNAs on interferon production. I now support publication.

Reviewer #3 (Remarks to the Author):

This reviewer has no further comment.